# Anterior Approach to Hip Arthroplasty with Early Mobilization Key for Reduced Hospital Length of Stay

**DOI:** 10.3390/medicina59071216

**Published:** 2023-06-28

**Authors:** Mihaela Bontea, Erika Bimbo-Szuhai, Iulia Codruta Macovei, Paula Bianca Maghiar, Mircea Sandor, Mihai Botea, Dana Romanescu, Corina Beiusanu, Adriana Cacuci, Liliana Sachelarie, Anca Huniadi

**Affiliations:** 1Department of Morphological Disciplines, Faculty of Medicine and Pharmacy, University of Oradea, 1st December Square 10, 410073 Oradea, Romania; bontea.mihaela@yahoo.ro (M.B.); beiucorina@yahoo.com (C.B.); adrianacacuci@yahoo.com (A.C.); 2Pelican Hospital, Corneliu Coposu Street 2, 410450 Oradea, Romania; icmek69@gmail.com (I.C.M.); pao.badea@gmail.com (P.B.M.); drmob78@yahoo.com (M.B.); danamed2000@yahoo.com (D.R.); ancahuniadi@gmail.com (A.H.); 3Faculty of Medicine and Pharmacy, University of Oradea, 1st December Square 10, 410073 Oradea, Romania; comdrims75@gmail.com; 4Department of Medical Disciplines, Faculty of Medicine and Pharmacy, University of Oradea, 1st December Square 10, 410073 Oradea, Romania; 5Department of Prelinical Discipline, Apollonia University, 700511 Iasi, Romania

**Keywords:** arthroplasty, hip joint, mobilization

## Abstract

*Background and Objectives:* This study aimed to explore the preoperative factors related to early mobilization and length of stay (LOS) after total hip arthroplasty and the benefits of the anterior approach over the traditional lateral approach. *Materials and Methods:* Every patient benefits from information regarding details of the surgery approach, possible intra, and postoperative complications, post-operator medical care, and steps in the early mobilization protocol. The patient underwent a pre-anesthetic evaluation, was checked for preoperatory vital function, and was reevaluated for mobilization at 6, 12, 24, 36, 48, and 96 h after total hip arthroplasty using the anterior versus lateral approach. *Results:* The result of the statistical calculations indicates the independent negative risk factors for reaching the mobilization target: age with a coefficient of −0.046, *p* = 0.0154 and lateral approach with a relative risk of 0.3802 (95% CI: 0.15–0.90), *p* = 0.0298. Statistical data concerning the length of stay (LOS) showed significant differences in the total days spent in the hospital. The patients who were operated on using the lateral approach presented a higher body mass index than those with the anterior approach, but this difference did not reach the threshold of statistical significance. *Conclusions:* In our study, patient mobilization is crucial to reduce LOS.

## 1. Introduction

Total hip arthroplasty (THA) is an important short-term application of biomaterials to alleviate pain, restore joints, and increase functional mobility in diseased traumatized joints [1]. The need for porous and lightweight hip implants due to the problems that arise makes it important to implement the best possible treatment for patients. The implementation of porous and lightweight hip implants during surgery solves most problems by facilitating better osseointegration, better medullary revascularization, and improved blood circulation in patients [1,2,3]. The surgical approach for THA, such as anterior or lateral, may affect the patient’s postoperative mobility and length of hospital stay.

The anterior approach for THA involves making an incision in the front of the hip joint, which allows less disruption of the surrounding muscles and a faster recovery time [4].

Total hip arthroplasty (THA) has become a standard surgical procedure for patients with hip disease [1]. Considered the “father of total hip arthroplasty”, Sir John Charnley’s pioneering work laid the foundation for modern hip arthroplasty, leading to greatly improved clinical outcomes. His low-friction arthroplasty technique using metal and polyethylene components and the use of cement for fixation revolutionized hip replacement surgery and greatly improved implant survival [2]. Surgical planning and surgical techniques have advanced significantly since Charnley’s practice. Implant design, materials, and perioperative care have improved patient outcomes [1,3]. Numerous studies have demonstrated the efficacy and safety of THA in reducing pain, improving function, and improving quality of life in patients with hip disease [4,5,6].

For a longer lifetime in a total hip prosthesis, the safety of the bearing must be a priority. Ensuring the safety of medical implants can be done both clinically and experimentally, but both require a relatively long time and are not cheap, so computational simulation-based studies appear to be a reasonable option [3]. The direct anterior hip approach for total hip arthroplasty is believed to have several advantages over other common approaches because it takes advantage of the natural intramuscular and intraneural gaps. Recent emphasis on minimally invasive, tissue-sparing ambulatory arthroplasty has significantly increased the use of direct anterior total hip arthroplasty. Proponents of this approach cite faster recovery times, less pain, improved patient satisfaction, and greater accuracy in implant placement and alignment and restoration of leg length [1,2].

Patients who undergo ATH with an anterior approach experience less postoperative pain and are able to move more quickly than those who undergo a lateral approach. Studies have shown that patients undergoing ATH through an anterior approach may have shorter hospital stays than those undergoing a lateral approach [1,2,3].

In the lateral approach to total hip replacement, an incision is made on the side of the hip to allow a more direct view of the joint and easier access to the acetabulum; Figure 1. The lateral approach provides better visualization and more accurate placement of the prosthesis, but it may also result in greater postoperative pain and longer recovery time [4].

Patients who received PTH with an anterior approach had a significantly shorter hospital stay compared with those who received a lateral approach. Note that the length of hospital stay varies depending on the patient’s condition, such as: B. His general health and previous illnesses may vary. Patients who received ATH with a lateral approach had a significantly longer hospital stay compared to those who received an anterior approach [5].

Postoperative mobilization is an important aspect of total hip arthroplasty recovery. Early ambulation is getting out of bed and walking as soon as possible after surgery. This helps reduce the risk of complications such as deep vein thrombosis and pneumonia. Long-term immobilization increases the risk of complications such as deep vein thrombosis, pulmonary embolism, pneumonia, tenderness, and muscle weakness. Patients with impaired performance status have a higher incidence of early postoperative complications. Properties of the anterior approach, such as B. Soft-tissue-sparing properties, allow faster rehabilitation and early hospital discharge [5,6,7,8].

Implementing early mobilization after anterior approach hip arthroplasty requires a multidisciplinary collaboration involving healthcare professionals such as surgeons, physiotherapists, and nurses. The patient’s individual needs and abilities must be considered when developing a mobilization plan.

Early mobilization should begin as soon as possible after surgery, ideally within the first 24–48 h. Patients should be encouraged to perform gentle exercises such as ankle pumps, leg raises, and walking with assistance. The physiotherapist should gradually increase the level of activity as the patient’s strength and mobility improves [9,10,11].

It is important to educate patients about the need for early mobilization and its proven benefits. Patients should be encouraged to actively participate in the recovery process by asking questions and raising concerns. Before hip replacement surgery, patients tend to change their gait to reduce pain. This preoperative adjustment and changes in muscle mass contribute to postoperative muscle function. In a study evaluating functional muscle recovery after minimally invasive total hip replacement surgery, Ward et al. found that preoperative gait function was the most important predictor of gait function, and age was negatively correlated with gait speed. [9].

Several authors reported a difference in length of stay of less than 24 h between various surgical approaches [8,9,10,11,12,13,14]. Most studies report that the mean length of stay is between two and five days. Sebečič et al. [15] reported that patients whose THA was undertaken through an anterior approach were discharged from hospital a mean of two days earlier than those who were operated on using a lateral approach.

For THA, a review of the literature showed that there is evidence associating the total comorbidity burden to an increased LOS [16].

The objective of the present article is to investigate the potential correlations of preoperative and postoperative factors with the length of hospital stay in the orthopedic ward for subjects that underwent THA. Mobilization and preoperative patients are the main key to reducing LOS.

## 2. Materials and Methods

### 2.1. Data Collection

Our clinical prospective observational cohort study was conducted in the Orthopedic Department of Oradea Pelican Clinical Hospital between January 2022 and December 2022, and we selected 121 patients who underwent THA after a diagnosis of osteonecrosis of the femoral head or osteoarthritis of the hip joint (degenerative, secondary). The work flow diagram is in Figure 2.

Following the selection criteria, 121 patients were included in the study, of which 6 did not sign the informed consent form. Of the remaining 115 patients, 66 underwent hip arthroplasty through an anterior approach, and 49 through a lateral approach.

This study was conducted according to the guidelines of the Declaration of Helsinki and approved by the Ethics Committee of Pelican Hospital Oradea (nr.2591/15.12.2021), and in the case of identifying an eligible patient, we proceeded to present, and have them sign, an informed consent protocol.

Exclusion criteria: age under 18, inflammatory hip arthritis, periprosthetic joint infection, history of revision surgery, patients with special devices due to severe instability, anatomical deformity, and refusing participation in this study. We excluded cases of inflammatory arthritis of the hip because, in this way, we avoid the occurrence of the postoperative risks related to infection. With regard to periprosthetic joint infection, this condition is a contraindication for intervention with the purpose of prosthetics because the present infection is a favorable condition for postoperative infectious complications.

The patients were randomized and were split into two batches based on the type of surgical approach: anterior approach versus lateral approach. The patient underwent a pre-anesthesia evaluation, checked for preoperatory vital function and the reevaluation of it, and mobilization at 6, 12, 24, 36, 48, and 96 h after hip arthroplasty for graduate mobilization. All patients started postoperative exercises following the same rehabilitation protocol. Alongside this, bedside exercises (ankle pumps, quadriceps stretching, leg raising) were performed 0–6 h after the operation. Standing and walker ambulation was permitted on postoperative day 1 following the same protocol. The pain was controlled conforming to our pain-control protocol.

In our hospital, every patient benefitted from information regarding details of the surgery approach, possible intra- and postoperative complications, postoperative medical care, and steps in the early mobilization protocol.

In our study, we designed a mobilization score as follows: 0—without mobilization; 1—mobilization at the edge of the bed; 2—mobilization with framework and help; 3—mobilization with the framework and no help; 4—mobilization with two crutches; and 5—mobilization with one crutch.

### 2.2. Statistical Analysis

The medical statistics program MedCalc^®^ version 12.5.0.0 (MedCalc^®^ Software, Mariakerke, Belgium) was used to store the information entered on the study sheet in a database and to perform statistical calculations. The results of the statistical tests are presented by the probability of the “null” hypothesis (*p*); its value below 0.05 proves a statistically significant difference between the studied batches. Certain results will also be displayed in graphic form using the same statistical program.

Depending on the nature of the variable, different parametric (for variables with normal distribution) or non-parametric (for variables with asymmetric distribution) tests were used. For parametric tests, we used Student’s test (*t*-test) for independent groups and for the non-parametric tests, we used the Mann–Whitney test. Categorical variables were described by their absolute values and percentages, in brackets. They were studied using the following tests: the chi-square test with Yates’ correction for continuity in the case of 2 × 2 frequency tables and the simple chi-square test for the other types of frequency tables (3 × 2, 3 × 3, etc.).

## 3. Results

### 3.1. Baseline Demographic and Clinical Criteria for the Two Study Groups

The patients were randomized using the same type of non-cemented prosthesis with ceramic friction on cross-linked polyethylene.

Following the selection criteria, 121 patients were included in the study, of which 6 did not sign the informed consent. Of the remaining 115 patients, 66 underwent hip arthroplasty through an anterior approach, and 49 through a lateral approach. The two batches were compared from a demographic and clinical point of view to be sure the postoperative evolution was not due to differences between the characteristics of the groups. Advanced age and lateral approach are independently correlated with failure of early mobilization. Gradual postoperative mobilization was faster for patients undergoing surgery via the anterior approach.

The demographic and clinical criteria for the two study groups are outlined in Table 1. It can be observed that there are no significant statistical differences between the anterior and lateral approach (*p* > 0.001).

### 3.2. Statistical Characteristics

As we can see from the table above that the two groups were comparable both demographically and clinically at baseline. The patients operated on via the lateral approach presented a higher body mass index than those with the anterior approach, but this difference did not reach the threshold of statistical significance. The comorbidities, in the two groups, are approximately similar as incidence.

Statistical data concerning the length of stay (LOS) showed significant differences regarding the total days spent in the hospital (a 5 days average with an 4–5 interquartile range for the anterior approach vs. 5 days with 5–6 interquartile range for the lateral approach, *p* = 0.0312), but not for days spent in the intensive care unit (1 day average with an 1–1 interquartile range for the anterior approach vs. 1 day average with an 1–1 interquartile range for the lateral approach, *p* = 0.7733); Figure 3.

Gradual postoperative mobilization was faster for patients undergoing surgery via the anterior approach. The value results expressed by median values of the mobilization scores are reproduced in Table 2 and Figure 4.

We considered that reaching score 5, for mobilization at 96 h after surgery, represents a satisfactory early mobilization result, and this was achieved by 30 of the patients in the anterior approach group (46.2%) but only by 12 cases in the lateral approach group (25%). The number of patients treated via the anterior approach needed to gain a patient who mobilized early (number needed to treat—NNT) is 4727 (IC 95%: 2.58–28.13), *p* = 0.0306.

With this score as a target at 96 h postoperatively, we also built a logistic regression model in which we entered the following variables: age, body mass index, and type of approach. The result of the statistical calculations indicates independent negative risk factors for reaching the mobilization target: age with a coefficient of −0.046, *p* = 0.0154, and lateral approach with a relative risk of 0.3802 (95% CI: 0.15–0.90), *p* = 0.0298. Thus, advanced age and the lateral approach are independently correlated with failure of early mobilization.

## 4. Discussion

Rapid recovery during the postoperative period has become a key element, and it is considered as a synthesis of the most advanced surgical and medical practices [17,18,19]. M. Tauviqirrahman et al. demonstrated in an experimental study carried out on hip prostheses with dual mobility using a femoral head diameter of 32 mm with an inclination of 45 degrees on the acetabular cup that it is a better technical alternative to reduce the loading of the joint [20].

M.I. Ammarullah et al. stated in a computational model study that the Ti6AI4V-on-Ti6AI4V model is clearly superior to the other models of metal-on-metal prostheses [21]. In the study carried out by us, the prosthesis model used was the uncemented one with ceramic friction on cross-linked polyethylene. Our study did not focus on the differences in types of prostheses, each patient benefiting from the same model of prosthesis, but on early postoperative mobilization and the benefits of preoperative instruction.

R.U. Putra et al. showed in their study that in the case of cemented prostheses, the level of human physiological activity has a negative impact on their degradation [22].

In the opinion of the manufacturer of non-cemented prosthesis with ceramic friction on cross-linked polyethylene, such a prosthesis which has a life of over 20–25 years, since the friction between the ceramic ball installed on the femur and the acetabular cup of cross-linked polyethylene produces few particles, the interval of time until a possible revision of the prosthesis is very long. The survival of the prosthesis in time also depends a lot on the activity level of each individual patient. Early mobilization has evolved over the past two decades and has proven its efficacy in terms of reducing hospital stay, morbidity, and recovery without an increase in readmission [20]. To lower risk factors that could increase perioperative complications or the possibility of new readmittance within 3 months, a multidisciplinary approach related to the management of organizational aspects and patient education had to be accurately addressed before the surgery. In our study, all patients who underwent a total hip arthroplasty using cementless prosthesis with ceramic friction on cross-linked polyethylene, and who met the inclusion criteria, were trained preoperatively regarding the recovery of joint mobility postoperatively. P.A. Vendittoli et al. in their ERAS short-stay protocol for THA managed to improve patient evolution by reducing the complication rate by 15% versus standard protocol. Women had longer stays than men and patients living alone stayed longer, which has also been shown in most studies [23,24,25,26]. There is no clear-cut explanation for this finding. In our study, in accordance with orthopedic protocol in use in our hospital, all patients without postoperative complications are discharged 3–5 days after surgery, and we did not find any differences regarding gender-related LOS.

The preoperative use of walking aids was associated with increased LOS. One possible explanation is differences in muscular strength and gait pattern between patients with and without walking aids [27,28]. Postoperatory mobilization is crucial in the reduced LOS, and our study proves that the lateral approach extends the hospitalization period compared to the anterior approach (*p* = 0.0237). Physiotherapy in our mobilization protocol was performed twice a day for the following 2 days after surgery. On the third day, if the program is respected, the patient is discharged without crutches reaching a full weight-bearing resumption. After discharge, patients in both surgical approaches are invited to continue home exercises to increase the range of motion of their hip, muscular strengthening, and proprioceptive recovery, but further rehabilitation sessions are not required.

Providing sufficient pain management is crucial for early rehabilitation and physical therapy, which is important for achieving an adequate range of motion of the joint and discharge from the hospital [29,30]. Controlling pain with our postoperative analgesic management protocol, we obtained early mobilization. As a consequence, we reduced LOS, and we observed slightly longer hospitalization in the lateral approach group (*p* = 0.7733).

Yoon et al. [30] showed that appropriate preoperative information reduces LOS. Therefore, in our study, during the pre-admission visit, a detailed multidisciplinary interview with an anesthesiologist, a surgeon, and other professional figures such as psychologists proposes to provide information on anesthesia and surgical techniques to minimize the degree of anxiety related to surgery and reducing pain during and after the surgical procedure. The physiotherapist provides information regarding exercises to be performed in the postoperatory period for early mobilization. Furthermore, it is demonstrated that the presence of chronic preoperative pain is linked to a greater probability of developing or maintaining pain in the postoperative period [31,32,33,34,35]; common risk factors (e.g., cardiovascular diseases, stroke, obesity, rheumatic diseases, type 2 diabetes mellitus, and anemia) must be carefully assessed before the surgery. Our protocol includes the multidisciplinary assessment of the patient in order to reduce possible intra- and postoperative complications. Some controversy still exists when evaluating the effect of obesity on the clinical outcomes of THA. Russo et al. [36] investigated differences in anterior approach complications among three groups (BMI < 25, BMI = 25 to 29.9 and BMI ≥ 30). Their results showed an increased risk of wounds and major complications, which were not significant, except for the increased length of hospital stay in the obese group.

In a study conducted by M.I. Ammarullah et al., the authors discovered that the body mass index is a factor that should be decisive from the choice of material to the geometry of the implant and to the type of technique adapted to obese patients [21].

In our study, in the case of the anterior approach, the average BMI was 28.5, respectively, and in the case of the lateral approach, we had a BMI of 29.9, so we can say that there is no statistically significant difference between the two groups. For our study group, BMI did not influence the decision of the surgical approach and, as a consequence, we did not find any statistical difference between groups (*p* = 0.1333).

Some studies have shown that obese patients have higher perioperative complications, more anesthetic complications, longer LOS, and poorer wound healing [33]. Furthermore, obesity is associated with poor short-term outcomes after undergoing THA, with reported complications of increased rates of periprosthetic infections, dislocations, and instability [37,38,39]. The clinical and functional outcomes of morbidly obese patients following primary THA remain unclear. Some studies, as well as our study, reported no significant difference in postoperative outcomes between obese and non-obese patients [39,40,41,42]. Despite this, it is common practice for patients with a BMI ≥ 40 kg/m^2^ to be asked to provide evidence of weight loss preoperatively [36]. Morbid obesity, in part, can also be a consequence of the limitation of mobilization due to chronic joint pain. The main purpose of total hip replacement is, in addition to regaining mobility, the cancellation of chronic pain, a fact that could be a decisive factor in resuming physical activity and, implicitly, weight loss.

The best functional outcomes and prosthesis survival rates after hip arthroplasty were reported among patients between 45 and 75 years of age [41]. In our study, in the case of the anterior approach, the average age of the group was 62.9 years, and in the case of the lateral approach, it was 65.3 years without reaching the level of statistical significance (*p* = 0.3080).

We consider that the limitations of the current study are the existence of only one operating team that applied the technique of the anterior approach versus the lateral approach, the period of the study, and the moderate number of patients undergoing prosthesis through via anterior approach, who mobilized early postoperatively. Using the same inclusion and exclusion criteria in the selection of patients, the two study groups had similar characteristics, with no significant differences except for the results of early mobilization in the case of patients operated on via the anterior approach.

THA is an established treatment option for patients with hip arthritis and other hip diseases. The future in implant technology, materials, surgical techniques, and perioperative management strategies shows better outcomes and patient satisfaction. The outlook for hip replacements includes continued improvements in implant materials, design, and surgical techniques to further improve implant performance and long-term durability. The individual adaptation of implants and the use of new technologies such as 3D printing and artificial intelligence are expected to optimize preoperative planning and implant selection. A number of issues, such as the increasing prevalence of obesity and comorbidities in young and elderly patients undergoing ATH, as well as long-term monitoring and optimization of patient care, need to be addressed. THA is a promising new field to further improve the quality of life of patients with hip disease.

## 5. Conclusions

In conclusion, we can state that in our group, advanced age and lateral approach are independently correlated with the failure of early mobilization. We did not find any differences in our study regarding gender-related LOS. Mobilization is crucial in the reduced LOS. By controlling pain with our postoperative analgesic management protocol, we obtained early mobilization and, as a consequence, we reduced LOS. BMI did not influence the decision of the surgical approach.

## Figures and Tables

**Figure 1 medicina-59-01216-f001:**
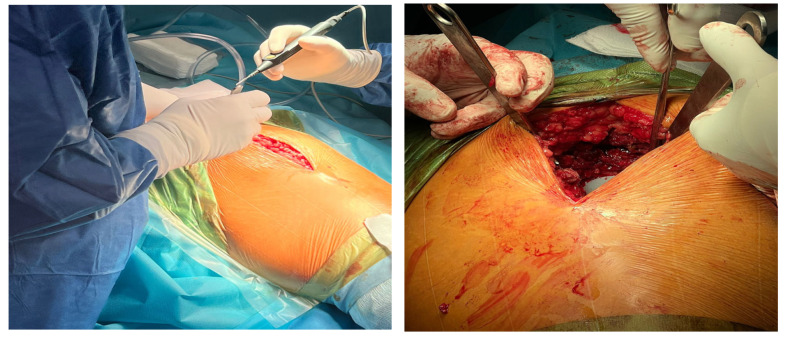
Incision on the side of the hip.

**Figure 2 medicina-59-01216-f002:**
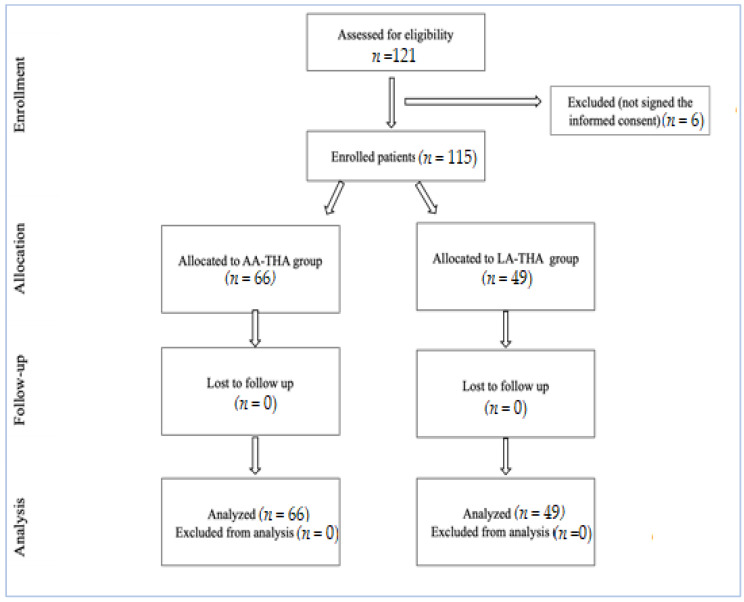
Consort flow diagram.

**Figure 3 medicina-59-01216-f003:**
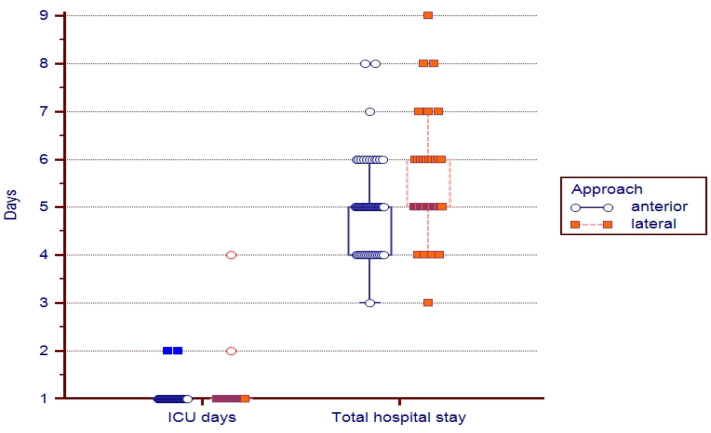
Median, 25th–75th percentile, and extreme values for hospital and intensive care days (ICU) for patients operated on via the anterior and lateral approach.

**Figure 4 medicina-59-01216-f004:**
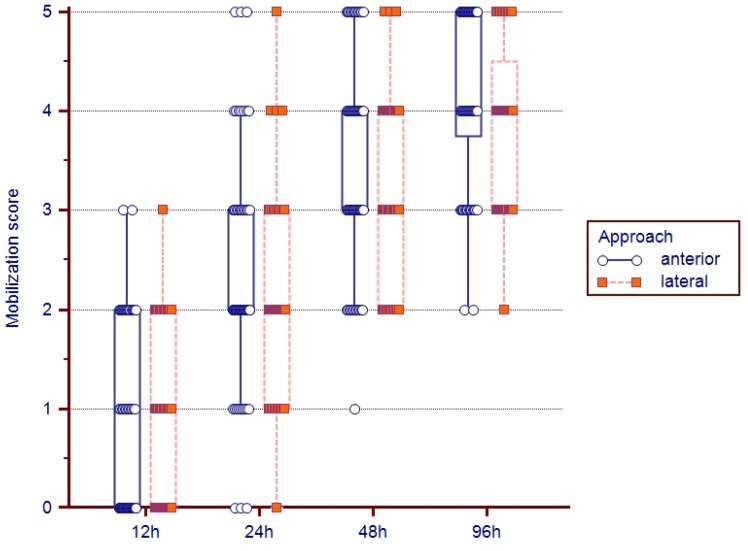
Evolution of postoperative mobilization scores (median, 25–75th percentile, and extreme values) for patients operated on via the anterior and lateral approach.

**Table 1 medicina-59-01216-t001:** Baseline demographic and clinical criteria for the two study groups.

	Anterior Approach (*n* = 66)	Lateral Approach (*n* = 49)	Statistical Significance (*p*)
Gender (M/F)	32/34	19/30	0.3972
Age—media (SD)	62.9 (12.5)	65.3 (10.7)	0.3080
Environment of origin (U/R)	49/17	32/17	0.4055
BMI (kg/m^2^)—media (SD)	28.5 (3.9)	29.9 (5.7)	0.1333
Comorbidities (percentage—%):			0.4513
HTN	43 (65.2)	30 (61.2)	
CHD	1 (1.5)	2 (4.1)	
MI	0 (0)	1 (2.0)	
CVA	0 (0)	2 (4.1)	
DM	13 (19.7)	10 (20.4)	
RhD	1 (1.5)	1 (2.0)	

M = male, F = female, SD = standard deviation, U = urban, R = rural, BMI = body mass index, HTN = hypertension, CHD = coronary heart disease, MI = previous myocardial infarction, CVA = cerebrovascular accident, DM = diabetes mellitus, RhD = rheumatic disease.

**Table 2 medicina-59-01216-t002:** Comparison of study groups in terms of postoperative mobilization.

	Anterior Approach (*n* = 66)	Lateral Approach (*n* = 49)	Statistical Significance (*p*)
Postoperative mobilization score 12 h—median (IQR)	0 (0–2)	1 (0–2)	0.4286
Postoperative mobilization score 24 h—median (IQR)	2 (2–3)	2 (1–3)	0.3406
Postoperative mobilization score 48 h—median (IQR)	3 (3–4)	3 (2–4)	0.2196
Postoperative mobilization score 96 h—median (IQR)	4 (3.75–5)	4 (3–4)	0.0237

IQR = interquartile range; mobilization score: 0—without mobilization, 1—mobilization at the edge of the bed, 2—mobilization with framework and help, 3—mobilization with framework and no help, 4—mobilization with two crutches, 5—mobilization with one crutch.

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
