# Peer review of "Anterior Approach to Hip Arthroplasty with Early Mobilization Key for Reduced Hospital Length of Stay"

_medicina, 2023, doi:10.3390/medicina59071216_

Round 1

Reviewer 1 Report

1.      In the abstract section, quantitative findings should be reported.

2.      Please give a "take-home" message as the conclusion of your abstract.

3.      Keywords should have been reorganized alphabetically.

4.      The Reviewer does not see the novel in the present article. My examination revealed that several similar previous publications appear to appropriately address the issues you have brought up in the current submission. Please emphasize it more advance in the introduction section if there are any more truly something really new.

5.      The work, novelty, and constraints of relevant previous literature must be explained in the introduction section to highlight the article gaps that the present work aims to fill.

6.      The end of a paragraph in the introduction section should explain the objective of the present article, the present form was not.

7.      Please provide an additional figure in the introduction section in related submission work to improve the reader's understanding.

8.      In the materials and methods, the authors need to add additional illustrations as a form of figure that explains the workflow of the present study to make the reader easier to understand rather than only the dominant text as a present form.

9.      Users of a hip arthroplasty perform most common of their activities by walking normally. The authors encouraged to include the explanation along with the reference as follows, doi: 10.1038/s41598-023-30725-6, 10.3390/ma14247554, and 10.3390/su15010823

10.   The authors must discuss the basis for patient selection because the current format is inappropriate. The quantity and diversity of the participating patients are extremely low, and there is no true group control. Please take this matter seriously as it is one of the main problems in the present submissions and the basis for suggesting a rejection.

11.   It's also important to provide more particular information on tools, such as the manufacturer, the country, and the specification.

12.   More serious discussion related to hip arthroplasty and body mass index needs to included. To address this issue, please refer and adopt suggested reference as follows: Tresca Stress Study of CoCrMo-on-CoCrMo Bearings Based on Body Mass Index Using 2D Computational Model. Jurnal Tribologi 2022, 33, 31–8. https://jurnaltribologi.mytribos.org/v33/JT-33-31-38.pdf

13.   The inaccuracy and tolerance of the experimental equipment used in this inquiry are critical details that must be included in the article.

14.   An evaluation of the findings with similar past research is essential.

15.   The discussion in the present article is extremely poor in quality overall. The authors must elaborate on their arguments and provide a thorough justification.

16.   Potential further study performing computational simulation/in silico in hip arthroplasty related study needs to be discussed. It brings several advantages compared to clinical study such as lower cost, faster results, and does not involved patient. The methods also would become preliminary study before performing clinical study. Also, relevant reference needs to included to support the explanation as follows, doi: 10.1016/j.matpr.2022.02.055, 10.3390/jfb12020038, and 10.3390/biomedicines11030951

17.   What is the current work's limitation? Please place it before entering the conclusion section.

18.   Further research should indeed be mentioned in the conclusion section.

19.   Sometimes in the entire manuscript, the authors created paragraphs with just one or two phrases, which made the explanation difficult to understand. To make their explanation a full paragraph, the authors must expand it. It is advised to use at least three sentences in a paragraph, with one acting as the primary sentence and the other two as supporting phrases.

20.   Literature from the last five years should be enriched to reference. MDPI reference is strongly recommended.

21.   I am recommending to the authors for reducing their level of self-citation in the current submission.

22.   English needs to be proofread due to grammatical errors and English style.

23.   Provide graphical abstract for submission after revision.

-

Author Response

The authors acknowledge the useful observations and suggestions of the reviewer’s as concerns the manuscript entitled

Anterior approach to hip arthroplasty with early mobilization key for reduced hospital length of stay

Mihaela Bontea 1, Erika Bimbo-Szuhai 1,3,*, Iulia Codruta Macovei 2,3, Paula Bianca Maghiar 2,3 , Mircea Sandor 2, Mihai Botea 2,3, Dana Romanescu 3,4, Corina Beiusanu1, Adriana Cacuci 1, Liliana Sachelarie 5,* and Anca Huniadi 2,3

According to the reviewer’s recommendations, the suggestions were carefully considered, as follows:

  1. In the abstract section, quantitative findings should be reported.

Done

  1. Please give a "take-home" message as the conclusion of your abstract.

Done

  1. Keywords should have been reorganized alphabetically.

Done

  1. The Reviewer does not see the novel in the present article. My examination revealed that several similar previous publications appear to appropriately address the issues you have brought up in the current submission. Please emphasize it more advance in the introduction section if there are any more truly something really new.

The novelty of the study carried out in our unit is represented by the aspects related to the introduction of the new anterior approach in hip arthroplasty. We started to practice the previous approach technique in September 2021, we developed a study sheet for the in-hospital monitoring of patient mobilization and implemented the study for the period 01.2022-12.2022. In our country, we are among the first clinics to introduce this approach technique.

  1. The work, novelty, and constraints of relevant previous literature must be explained in the introduction section to highlight the article gaps that the present work aims to fill.

The present paper wants to highlight the results obtained during the learning period of the new previous approach technique. The data from the specialized literature are richer in relation to the comparison between the anterior and the posterior approach, providing more data on the details of the technique and less on the postoperative management regarding the mobilization of the patient.

  1. The end of a paragraph in the introduction section should explain the objective of the present article, the present form was not.

We have modified the objective of the article to be more obvious.

  1. Please provide an additional figure in the introduction section in related submission work to improve the reader's understanding.

Done

  1. In the materials and methods, the authors need to add additional illustrations as a form of figure that explains the workflow of the present study to make the reader easier to understand rather than only the dominant text as a present form.

We introduced flow consort according to the requirements of mdpi. Materials and methods: All patients benefited from the same type of prosthesis: non-cemented prosthesis with ceramic friction on cross-linked polyethylene De Puy Johnson&Johnson with four components.

  1. Users of a hip arthroplasty perform most common of their activities by walking normally. The authors encouraged to include the explanation along with the reference as follows, doi: 10.1038/s41598-023-30725-6, 10.3390/ma14247554, and 10.3390/su15010823
  2. Tauviqirrahman et. all demonstrated in an experimental study carried out on hip prostheses with dual mobility using a femoral head diameter of 32mm with an inclination of 45 degrees on the acetabular cup, it is a better technical alternative to reduce the loading of the joint. (two: 10.1038/s41598-023-30725-6).

M.I.Ammarullah et all specify in a computational model study that the Ti6AI4V-on-Ti6AI4V model is clearly superior to the other models of metal-on-metal prostheses (10.3390/ma14247554). In the study carried out by us, the prosthesis model used was the uncemented one with ceramic friction on cross-linked polyethylene. Our study did not focus on the differences in the types of prostheses, each patient benefiting from the same prosthesis model, but on early postoperative mobilization and the benefits preoperative instruction.

R.U. Putra et. all showed in a study that in the case of cemented prostheses, the level of human physiological activity has a negative impact on their degradation (10.3390/su15010823).

In the opinion of the manufacturer of non-cemented prosthesis with ceramic friction on cross-linked polyethylene, such a prosthesis has a life of over 20-25 years, since the friction between the ceramic ball installed on the femur and the acetabular cup of cross-linked polyethylene produces few particles, the interval of time until a possible revision of the prosthesis being very long. The survival of the prosthesis in time also depends a lot on the activity level of each individual patient.10.   The authors must discuss the basis for patient selection because the current format is inappropriate. The quantity and diversity of the participating patients are extremely low, and there is no true group control. Please take this matter seriously as it is one of the main problems in the present submissions and the basis for suggesting a rejection.

  1. It's also important to provide more particular information on tools, such as the manufacturer, the country, and the specification.

The type of prosthesis used is an uncemented prosthesis with ceramic friction on cross-linked polyethylene, from the manufacturer Johnson & Johnson, De Puy, manufactured in the USA.

  1. More serious discussion related to hip arthroplasty and body mass index needs to included. To address this issue, please refer and adopt suggested reference as follows: Tresca Stress Study of CoCrMo-on-CoCrMo Bearings Based on Body Mass Index Using 2D Computational Model. Jurnal Tribologi 2022, 33, 31–8. https://jurnaltribologi.mytribos.org/v33/JT-33-31-38.pdf

In a study conducted by M.I. Ammarullah et. all the authors discovered that the body mass index is a factor that should be decisive from the choice of material, to the geometry of the implant, to the type of technique adapted to obese patients. (https://jurnaltribologi.mytribos.org/v33/JT-33-31-38.pdf).

In our study, in the case of the anterior approach, the average BMI was 28.5, respectively, in the case of the lateral approach, we had a BMI of 29.9, so we can say that there is no statistically significant difference between the two groups.

  1. The inaccuracy and tolerance of the experimental equipment used in this inquiry are critical details that must be included in the article.

We did not use experimental equipment we only used existing prostheses on the market.

  1. An evaluation of the findings with similar past research is essential.

The data from the specialized literature comparing the two techniques of the anterior versus lateral approach (during the implementation period of the anterior approach technique) on a single type of prosthesis with the early postoperative results related to the mobilization and length of hospitalization are somewhat deficient in our country.

  1. The discussion in the present article is extremely poor in quality overall. The authors must elaborate on their arguments and provide a thorough justification.

Done

  1. Potential further study performing computational simulation/in silico in hip arthroplasty related study needs to be discussed. It brings several advantages compared to clinical study such as lower cost, faster results, and does not involved patient. The methods also would become preliminary study before performing clinical study. Also, relevant reference needs to included to support the explanation as follows, doi: 10.1016/j.matpr.2022.02.055, 10.3390/jfb12020038, and 10.3390/biomedicines11030951

We apologize, but regarding the links recommended to be studied, we have a confusion as to how we could include computational simulation models in the discussions regarding the clinical results on our lot.

  1. What is the current work's limitation? Please place it before entering the conclusion section.

We consider that the limitation of the current study is given by the existence of only one operating team that applied the technique of the anterior approach versus the lateral approach, the period of the study and the moderate number of patients undergoing prosthesis through the anterior approach, mobilized early postoperatively. Using the same inclusion and exclusion criteria in the selection of patients, the two study groups had similar characteristics, with no significant differences except for the results of early mobilization in the case of patients operated through the anterior approach.

  1. Further research should indeed be mentioned in the conclusion section.

Additional research would be welcome with the comparison of the results of several teams of orthopedics, a larger number of enrolled patients, a longer period of monitoring of the team, on several types of prostheses and the comparison of the results between cemented and uncemented ones with an early mobilization protocol patients.

  1. Sometimes in the entire manuscript, the authors created paragraphs with just one or two phrases, which made the explanation difficult to understand. To make their explanation a full paragraph, the authors must expand it. It is advised to use at least three sentences in a paragraph, with one acting as the primary sentence and the other two as supporting phrases.

Done

  1. Literature from the last five years should be enriched to reference. MDPI reference is strongly recommended.

Done

  1. I am recommending to the authors for reducing their level of self-citation in the current submission.

Done

  1. English needs to be proofread due to grammatical errors and English style.

Done

  1. Provide graphical abstract for submission after revision.

Will provide.

Thank you very much for review reports and for the extremely useful observations and suggestions!

Kind regards,

Prof.dr. Liliana Sachelarie

Reviewer 2 Report

The purpose of this study was to investigate the preoperative factors associated with early mobilization and longer length of stay after total hip arthroplasty, as well as the advantages of the anterior approach over the traditional lateral approach.

A considerable amount of literature has already been published on this subject. The introduction section should include an explanation of the previous literature's work, novelty, and constraints that are relevant to the present work. This will help to highlight the gaps in the article that the present work aims to fill.

To enhance the reader's comprehension of the present study, the authors should consider including additional figures in the materials and methods section that illustrate the workflow. 

The main message of the study could be highlighted a bit better.

 The reference list should be updated to include literature from the last five years.

Tables and Figures are ok.

Ok

Author Response

The authors acknowledge the useful observations and suggestions of the reviewer’s as concerns the manuscript entitled

Anterior approach to hip arthroplasty with early mobilization key for reduced hospital length of stay

Mihaela Bontea 1, Erika Bimbo-Szuhai 1,3,*, Iulia Codruta Macovei 2,3, Paula Bianca Maghiar 2,3 , Mircea Sandor 2, Mihai Botea 2,3, Dana Romanescu 3,4, Corina Beiusanu1, Adriana Cacuci 1, Liliana Sachelarie 5,* and Anca Huniadi 2,3

According to the reviewer’s recommendations, the suggestions were carefully considered, as follows:

The purpose of this study was to investigate the preoperative factors associated with early mobilization and longer length of stay after total hip arthroplasty, as well as the advantages of the anterior approach over the traditional lateral approach.

The present paper wants to highlight the results obtained during the learning period of the new previous approach technique. The data from the specialized literature are richer in relation to the comparison between the anterior and the posterior approach, providing more data on the details of the technique and less on the postoperative management regarding the mobilization of the patient. We consider that the limitation of the current study is given by the limited existence of a single team that applied this anterior approach versus the lateral one, the period of the study and the moderate number of patients undergoing prosthesis through the anterior approach, mobilized early postoperatively.

A considerable amount of literature has already been published on this subject. The introduction section should include an explanation of the previous literature's work, novelty, and constraints that are relevant to the present work. This will help to highlight the gaps in the article that the present work aims to fill.

The data from the specialized literature comparing the two techniques of the anterior versus lateral approach (during the implementation period of the anterior approach technique) on a single type of prosthesis with the early postoperative results related to the mobilization and length of hospitalization are somewhat deficient in our country.

To enhance the reader's comprehension of the present study, the authors should consider including additional figures in the materials and methods section that illustrate the workflow. 

Done

The main message of the study could be highlighted a bit better.

In our study patient mobilization is crucial in the reduced LOS.  

 The reference list should be updated to include literature from the last five years.

Done

Tables and Figures are ok.

Thank you very much for review reports and for the extremely useful observations and suggestions!

Kind regards,

Prof.dr. Liliana Sachelarie

Reviewer 3 Report

1.  Authors described that “We split the group into two batches depending on the type of surgical approach”in line 120. How did the author split the two groups? Authors should state in detail whether they were randomized or not.

2.  Authors described thatThe patients operated through lateral approach presented a higher body mass index than those with anterior approach, but this difference did not reach the threshold of statistical significance”in line 165-6. I think lateral approach also presented advanced age. Authors should be age-matched using the Propensity Score Matching Methods, and reconsidered statistics.

Author Response

The authors acknowledge the useful observations and suggestions of the reviewer’s as concerns the manuscript entitled

Anterior approach to hip arthroplasty with early mobilization key for reduced hospital length of stay

Mihaela Bontea 1, Erika Bimbo-Szuhai 1,3,*, Iulia Codruta Macovei 2,3, Paula Bianca Maghiar 2,3 , Mircea Sandor 2, Mihai Botea 2,3, Dana Romanescu 3,4, Corina Beiusanu1, Adriana Cacuci 1, Liliana Sachelarie 5,* and Anca Huniadi 2,3

According to the reviewer’s recommendations, the suggestions were carefully considered, as follows:

Authors described that “We split the group into two batches depending on the type of surgical approach”in line 120. How did the author split the two groups? Authors should state in detail whether they were randomized or not.

 The patients were randomized, using the same type of non-cemented prosthesis with ceramic friction on cross-linked polyethylene.

  1. Authors described that“The patients operated through lateral approach presented a higher body mass index than those with anterior approach, but this difference did not reach the threshold of statistical significance”in line 165-6. I think lateral approach also presented advanced age. Authors should be age-matched using the Propensity Score Matching Methods, and reconsidered statistics.

In our study, in the case of the anterior approach, the average BMI was 28.5, respectively, in the case of the lateral approach, we had a BMI of 29.9, so we can say that there is no statistically significant difference between the two groups.

In our study, in the case of the anterior approach, the average age of the group was 62.9 years, and in the case of the lateral approach, it was 65.3 years without reaching the level of statistical significance (p=0.3080).

Thank you very much for review reports and for the extremely useful observations and suggestions!

Kind regards,

Prof.dr. Liliana Sachelarie

Round 2

Reviewer 1 Report

Well effort so far, some correction and improvement still needed.

1.      Please enhance the novel of present manuscript since the novel is still weak.

2.      In the introduction section, related to anterior and lateral approach would be nice to provide illustration for this surgery to improve authors understanding.

3.      In line 34-36, the authors use [1-3] that out of the date not from last 5 years and use it several times in the article, please use updated literature: https://doi.org/10.3390/ma16093298, https://doi.org/10.3390/biomedicines11030951, and https://doi.org/10.3390/jfb12020038

4.      In line 47-48, related to this explanation, please to add/replace with relevant reference as follows: https://doi.org/10.1016/j.matpr.2022.02.055, https://doi.org/10.1088/1742-6596/1198/4/042012, and https://doi.org/10.3390/su142013413

5.      In line 120, where is the number of ethics? Just date?

6.      In line 123, why the exclusion criteria given on that? Please explain.

7.      In line 169, in depth table description is needed.

-

Author Response

The authors acknowledge the useful observations and suggestions of the reviewer’s as concerns the manuscript entitled

Anterior approach to hip arthroplasty with early mobilization key for reduced hospital length of stay

Mihaela Bontea 1, Erika Bimbo-Szuhai 1,3,*, Iulia Codruta Macovei 2,3, Paula Bianca Maghiar 2,3 , Mircea Sandor 2, Mihai Botea 2,3, Dana Romanescu 3,4, Corina Beiusanu1, Adriana Cacuci 1, Liliana Sachelarie 5,* and Anca Huniadi 2,3

According to the reviewer’s recommendations, the suggestions were carefully considered, as follows:

  1. Please enhance the novel of present manuscript since the novel is still weak.

Done

  1. In the introduction section, related to anterior and lateral approach would be nice to provide illustration for this surgery to improve authors understanding.

Figure 1 Incision on the side of the hip

  1. In line 34-36, the authors use [1-3] that out of the date not from last 5 years and use it several times in the article, please use updated literature: https://doi.org/10.3390/ma16093298, https://doi.org/10.3390/biomedicines11030951, and https://doi.org/10.3390/jfb12020038

  1. Jamari, J.; Ammarullah, M.I.; Saad, A.P. M.; Syahrom, A.; Uddin, ; Van der Heide, E.; Basri Hasan, The Effect of Bottom Profile Dimples on the Femoral Head on Wear in Metal-on-Metal Total Hip Arthroplasty. J. Funct. Biomater. 202112(2), 38.
  2. Ammarullah, M.I.;, Harton, R.; Supriyon, T.; Santoso, G.; Sugiharto, S.; Permana, M.S. Polycrystalline Diamond as a Potential Material for the Hard-on-Hard Bearing of Total Hip Prosthesis: Von Mises Stress Analysis. Biomedicines 2023, 11(3), 951.
  3. Salaha, Z. F.M.; Ammarullah, M. I.; Abdullah, N.N. A. A.; Aziz, A.U. A.; Gan, H.S.; Abdul Abdullah, H.; Kadir, M.R. A.; Ramlee, M. H. Biomechanical Effects of the Porous Structure of Gyroid and Voronoi Hip Implants: A Finite Element Analysis Using an Experimentally Validated Model. Materials 2023, 16(9), 3298.

  1. In line 47-48, related to this explanation, please to add/replace with relevant reference as follows: https://doi.org/10.1016/j.matpr.2022.02.055, https://doi.org/10.1088/1742-6596/1198/4/042012, and https://doi.org/10.3390/su142013413

Ammarullah, M. I.; Santoso, .; Sugiharto, S.; Supriyono, T.; Wibowo,D. B.; Kurdi, O.; Tauviqirrahman, M.; Jamari, J. Minimizing Risk of Failure from Ceramic-on-Ceramic Total Hip Prosthesis by Selecting Ceramic Materials Based on Tresca Stress.Sustainability 202214(20), 13413.

Ammarullah, M. I. I.; Afif, Y.; Maula, M. I.; Winarni, T. I.; Tauviqirrahman, M.; Jamari, J.Tresca stress evaluation of Metal-on-UHMWPE total hip arthroplasty during peak loading from normal walking activity Materialstoday2022, 63(1), S143-S146

Basri, Hasan; Syahrom, A; Prakoso, A T; Wicaksono, D; Amarullah, M I; et al.  The Analysis of Dimple Geometry on Artificial Hip Joint to the Performance of Lubrication, Journal of Physics: Conference Series; Bristol 2019,1198, 4. 

  1. In line 120, where is the number of ethics? Just date?

This study was conducted according to the guidelines of the Declaration of Helsinki, and approved by the Ethics Committee of Pelican Hospital Oradea (nr.2591/15.12.2021).

  1. In line 123, why the exclusion criteria given on that? Please explain.

In our orthopedics department, cases under the age of 18 are not treated, because pediatric orthopedics cases are treated in the pediatric orthopedics department.

We excluded cases of inflammatory arthritis of the hip because in this way we avoid the occurrence of postoperative risks related to infection.

With regard to periprosthetic joint infection, this condition is a contraindication for intervention with the purpose of prosthetics, because the present infection is a favorable condition for postoperative infectious complications.

The revision history of the prosthesis, patients with special devices, severe instability and anatomical deformations (fractures with vicious consolidation and congenital dislocation of the hip) require additional adaptation parts in the context of the prosthesis.

  1. In line 169, in depth table description is need

The demographic and clinical criteria for the two study groups are given in Table 1. It is observed that there are no significant statistical differences between anteriot and, lateral approach (p>0.001).

Thank you very much for review reports and for the extremely useful observations and suggestions!

Kind regards,

Prof.dr. Liliana Sachelarie

Reviewer 2 Report

All points were taken into account during the revision.

Author Response

The authors acknowledge the useful observations and suggestions of the reviewer’s as concerns the manuscript entitled

Anterior approach to hip arthroplasty with early mobilization key for reduced hospital length of stay

Mihaela Bontea 1, Erika Bimbo-Szuhai 1,3,*, Iulia Codruta Macovei 2,3, Paula Bianca Maghiar 2,3 , Mircea Sandor 2, Mihai Botea 2,3, Dana Romanescu 3,4, Corina Beiusanu1, Adriana Cacuci 1, Liliana Sachelarie 5,* and Anca Huniadi 2,3

Thank you very much for reviewing the reports and the extremely useful observations and suggestions!

Kind regards,

Prof. dr. Liliana Sachelarie

Reviewer 3 Report

1.  Authors described that “The patients were randomized, using the same type of non-cemented prosthesis with ceramic friction on cross-linked polyethylene.”in comments. However, there were not term of “randomized”in line 126-7. Authors should add “randomized”.

2.  Yoon et al.[31]--- in line 253. The reference number of Yoon et al. is “30”.

Author Response

The authors acknowledge the useful observations and suggestions of the reviewer’s as concerns the manuscript entitled

Anterior approach to hip arthroplasty with early mobilization key for reduced hospital length of stay

Mihaela Bontea 1, Erika Bimbo-Szuhai 1,3,*, Iulia Codruta Macovei 2,3, Paula Bianca Maghiar 2,3 , Mircea Sandor 2, Mihai Botea 2,3, Dana Romanescu 3,4, Corina Beiusanu1, Adriana Cacuci 1, Liliana Sachelarie 5,* and Anca Huniadi 2,3

According to the reviewer’s recommendations, the suggestions were carefully considered, as follows:

  1. Authors described that “The patients were randomized, using the same type of non-cemented prosthesis with ceramic friction on cross-linked polyethylene.”in comments. However, there were not term of “randomized”in line 126-7. Authors should add “randomized”.

 Done

  1. Yoon et al.[31]--- in line 253. The reference number of Yoon et al. is “30”.

Done

Thank you very much for review reports and for the extremely useful observations and suggestions!

Kind regards,

Prof.dr. Liliana Sachelarie
